# Policy Shaping: Integrating Human Feedback with Reinforcement Learning

Shane Griffith, Kaushik Subramanian, Jonathan Scholz, Charles L. Isbell, and Andrea Thomaz

College of Computing
Georgia Institute of Technology, Atlanta, GA 30332, USA
{sgriffith7, kausubbu, jkscholz}@gatech.edu,
{isbell, athomaz}@cc.gatech.edu

## Abstract

A long term goal of Interactive Reinforcement Learning is to incorporate non-expert human feedback to solve complex tasks. Some state-of-the-art methods have approached this problem by mapping human information to rewards and values and iterating over them to compute better control policies. In this paper we argue for an alternate, more effective characterization of human feedback: Policy Shaping. We introduce **Advise**, a Bayesian approach that attempts to maximize the information gained from human feedback by utilizing it as direct policy labels. We compare **Advise** to state-of-the-art approaches and show that it can outperform them and is robust to infrequent and inconsistent human feedback.

## 1 Introduction

A long–term goal of machine learning is to create systems that can be interactively trained or guided by non-expert end-users. This paper focuses specifically on integrating human feedback with Reinforcement Learning. One way to address this problem is to treat human feedback as a shaping reward [1–5]. Yet, recent papers have observed that a more effective use of human feedback is as direct information about policies [6, 7]. Most techniques for learning from human feedback still, however, convert feedback signals into a reward or a value. In this paper we introduce *Policy Shaping*, which formalizes the *meaning* of human feedback as policy feedback, and demonstrates how to use it directly as policy advice. We also introduce **Advise**, an algorithm for estimating a human's Bayes optimal feedback policy and a technique for combining this with the policy formed from the agent's direct experience in the environment (Bayesian Q-Learning).

We validate our approach using a series of experiments. These experiments use a simulated human teacher and allow us to systematically test performance under a variety of conditions of infrequent and inconsistent feedback. The results demonstrate two advantages of **Advise**: 1) it is able to outperform state of the art techniques for integrating human feedback with Reinforcement Learning; and 2) by formalizing human feedback, we avoid ad hoc parameter settings and are robust to infrequent and inconsistent feedback.

## 2 Reinforcement Learning

Reinforcement Learning (RL) defines a class of algorithms for solving problems modeled as a Markov Decision Process (MDP). An MDP is specified by the tuple $(S, A, T, R)$, which defines the set of possible world states, $S$, the set of actions available to the agent in each state, $A$, the transition function $T : S \times A \to \Pr[S]$, a reward function $R : S \times A \to \mathbb{R}$, and a discount factor $0 \leq \gamma \leq 1$. The goal of a Reinforcement Learning algorithm is to identify a policy, $\pi : S \to A$, which maximizes the expected reward from the environment. Thus, the reward function acts as a single source of information that tells an agent what is the best policy for this MDP.

This paper used an implementation of the Bayesian $Q$-learning (BQL) Reinforcement Learning algorithm [8], which is based on Watkins' $Q$-learning [9]. $Q$-learning is one way to find an optimal

policy from the environment reward signal. The policy for the whole state space is iteratively refined by dynamically updating a table of Q-values. A specific Q-value, $Q[s, a]$, represents a point estimate of the long-term expected discounted reward for taking action $a$ in state $s$.

Rather than keep a point estimate of the long-term discounted reward for each state-action pair, Bayesian $Q$-learning maintains parameters that specify a normal distribution with unknown mean and precision for each Q-value. This representation has the advantage that it approximates the agent's uncertainty in the optimality of each action, which makes the problem of optimizing the exploration/exploitation trade-off straightforward. Because the Normal-Gamma (NG) distribution is the conjugate prior for the normal distribution, the mean and the precision are estimated using a NG distribution with hyperparameters $\langle \mu_0^{s,a}, \lambda^{s,a}, \alpha^{s,a}, \beta^{s,a} \rangle$. These values are updated each time an agent performs an action $a$ in state $s$, accumulates reward $r$, and transitions to a new state $s'$. Details on how these parameters are updated can be found in [8]. Because BQL is known to under-explore, $\beta^{s,a}$ is updated as shown in [10] using an additional parameter $\theta$.

The NG distribution for each Q-value can be used to estimate the probability that each action $a \in A_s$ in a state $s$ is optimal, which defines a policy, $\pi_R$, used for action selection. The optimal action can be estimated by sampling each $\hat{Q}(s, a)$ and taking the argmax. A large number of samples can be used to approximate the probability an action is optimal by simply counting the number of times an action has the highest Q-value [8].

## 3 Related Work

A key feature of Reinforcement Learning is the use of a reward signal. The reward signal can be modified to suit the addition of a new information source (this is known as *reward shaping* [11]). This is the most common way human feedback has been applied to RL [1–5]. However, several difficulties arise when integrating human feedback signals that may be infrequent, or occasionally inconsistent with the optimal policy–violating the necessary and sufficient condition that a shaping function be potential-based [11]. Another difficulty is the ambiguity of translating a statement like "yes, that's right" or "no, that's wrong" into a reward. Typically, past attempts have been a manual process, yielding *ad hoc* approximations for specific domains. Researchers have also extended reward shaping to account for idiosyncrasies in human input. For example, adding a drift parameter to account for the human tendency to give less feedback over time [1, 12].

Advancements in recent work sidestep some of these issues by showing human feedback can instead be used as policy feedback. For example, Thomaz and Breazeal [6] added an *UNDO* function to the negative feedback signal, which forced an agent to backtrack to the previous state after its value update. Work by Knox and Stone [7, 13] has shown that a general improvement to learning from human feedback is possible if it is used to directly modify the action selection mechanism of the Reinforcement Learning algorithm. Although both approaches use human feedback to modify an agent's exploration policy, they still treat human feedback as either a reward or a value. In our work, we assume human feedback is not an evaluative reward, but is a label on the optimality of actions. Thus the human's feedback is making a direct statement about the policy itself, rather than influencing the policy through a reward.

In other works, rather than have the human input be a reward shaping input, the human provides demonstrations of the optimal policy. Several papers have shown how the policy information in human demonstrations can be used for inverse optimal control [14, 15], to seed an agent's exploration [16, 17], and in some cases be used entirely in place of exploration [18, 19]. Our work similarly focuses on people's knowledge of the policy, but instead of requiring demonstrations we want to allow people to simply critique the agent's behavior ("that was right/wrong").

Our position that human feedback be used as direct policy advice is related to work in transfer learning [20, 21], in which an agent learns with "advice" about how it should behave. This advice is provided as first order logic rules and is also provided offline, rather than interactively during learning. Our approach only requires very high-level feedback (right/wrong) and is provided interactively.

## 4 Policy Shaping

In this section, we formulate human feedback as policy advice, and derive a Bayes optimal algorithm for converting that feedback into a policy. We also describe how to combine the feedback policy with the policy of an underlying Reinforcement Learning algorithm. We call our approach **Advise**.

### 4.1 Model Parameters

We assume a scenario where the agent has access to communication from a human during its learning process. In addition to receiving environmental reward, the agent may receive a "right"/"wrong" label after performing an action. In related work, these labels are converted into shaping rewards (e.g., "right" becomes $+1$ and "wrong" $-1$), which are then used to modify Q-values, or to bias action selection. In contrast, we use this label directly to infer what the human believes is the optimal policy in the labeled state.

Using feedback in this way is not a trivial matter of pruning actions from the search tree. Feedback can be both inconsistent with the optimal policy and sparsely provided. Here, we assume a human providing feedback knows the right answer, but noise in the feedback channel introduces inconsistencies between what the human intends to communicate and what the agent observes. Thus, feedback is consistent, $\mathcal{C}$, with the optimal policy with probability $0 < \mathcal{C} < 1$.[1]

We also assume that a human watching an agent learn may not provide feedback after every single action, thus the likelihood, $\mathcal{L}$, of receiving feedback has probability $0 < \mathcal{L} < 1$. In the event feedback is received, it is interpreted as a comment on the optimality of the action just performed. The issue of credit assignment that naturally arises with learning from real human feedback is left for future work (see [13] for an implementation of credit assignment in a different framework for learning from human feedback).

### 4.2 Estimating a Policy from Feedback

It is possible that the human may know any number of different optimal actions in a state, the probability an action, $a$, in a particular state, $s$, is optimal is independent of what labels were provided to the other actions. Subsequently, the probability $s, a$ is optimal can be computed using only the "right" and "wrong" labels associated with it. We define $\Delta_{s,a}$ to be the difference between the number of "right" and "wrong" labels. The probability $s, a$ is optimal can be obtained using the binomial distribution as:

$$\frac{\mathcal{C}^{\Delta_{s,a}}}{\mathcal{C}^{\Delta_{s,a}} + (1-\mathcal{C})^{\Delta_{s,a}}} , \tag{1}$$

Although many different actions may be optimal in a given state, we will assume for this paper that the human knows only one optimal action, which is the one they intend to communicate. In that case, an action, $a$, is optimal in state $s$ if no other action is optimal (i.e., whether it is optimal now also depends on the labels to the other actions in the state). More formally:

$$\mathcal{C}^{\Delta_{s,a}}(1-\mathcal{C})^{\sum_{j \neq a} \Delta_{s,j}} \tag{2}$$

We take Equation 2 to be the probability of performing $s, a$ according to the feedback policy, $\pi_F$ (i.e., the value of $\pi_F(s, a)$). This is the Bayes optimal feedback policy given the "right" and "wrong" labels seen, the value for $\mathcal{C}$, and that only one action is optimal per state. This is obtained by application of Bayes' rule in conjunction with the binomial distribution and enforcing independence conditions arising from our assumption that there is only one optimal action. A detailed derivation of the above results is available in the Appendix Section A.1 and A.2.

### 4.3 Reconciling Policy Information from Multiple Sources

Because the use of **Advise** assumes an underlying Reinforcement Learning algorithm will also be used (e.g., here we use BQL), the policies derived from multiple information sources must be reconciled. Although there is a chance, $\mathcal{C}$, that a human could make a mistake when s/he does provide feedback, given sufficient time, with the likelihood of feedback, $\mathcal{L} > 0.0$ and the consistency of feedback $\mathcal{C} \neq 0.5$, the total amount of information received from the human should be enough for the the agent to choose the optimal policy with probability 1.0. Of course, an agent will also be learning on its own at the same time and therefore may converge to its own optimal policy much sooner than it learns the human's policy. Before an agent is completely confident in either policy, however, it has to determine what action to perform using the policy information each provides.

| Pac-Man | Frogger |
|---|---|

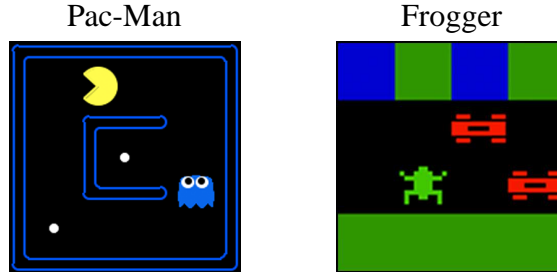

Figure 1: A snapshot of each domain used for the experiments. Pac-Man consisted of a 5x5 grid world with the yellow Pac-Man avatar, two white food pellets, and a blue ghost. Frogger consisted of a 4x4 grid world with the green Frogger avatar, two red cars, and two blue water hazards.

We combine the policies from multiple information sources by multiplying them together: $\pi \propto \pi_R \times \pi_F$. Multiplying distributions together is the Bayes optimal method for combining probabilities from (conditionally) independent sources [22], and has been used to solve other machine learning problems as well (e.g., [23]). Note that BQL can only approximately estimate the uncertainty that each action is optimal from the environment reward signal. Rather than use a different combination method to compensate for the fact that BQL converges too quickly, we introduced the exploration tuning parameter, $\theta$, from [10], that can be manually tuned until BQL performs close to optimal.

## 5    Experimental Setup

We evaluate our approach using two game domains, Pac-Man and Frogger (see Fig. 1).

### 5.1    Pac-Man

Pac-Man consists of a 2-D grid with food, walls, ghosts, and the Pac-Man avatar. The goal is to eat all the food pellets while avoiding moving ghosts (+500). Points are also awarded for each food pellet (+10). Points are taken away as time passes (-1) and for losing the game (-500). Our experiments used a $5 \times 5$ grid with two food pellets and one ghost. The action set consisted of the four primary cartesian directions. The state representation included Pac-Man's position, the position and orientation of the ghost and the presence of food pellets.

### 5.2    Frogger

Frogger consists of a 2-D map with moving cars, water hazards, and the Frogger avatar. The goal is to cross the road without being run over or jumping into a water hazard (+500). Points are lost as time passes (-1), for hopping into a water hazard (-500), and for being run over (-500). Each car drives one space per time step. The car placement and direction of motion is randomly determined at the start and does not change. As a car disappears off the end of the map it reemerges at the beginning of the road and continues to move in the same direction. The cars moved only in one direction, and they started out in random positions on the road. Each lane was limited to one car. Our experiments used a $4 \times 4$ grid with two water hazards and two cars. The action set consisted of the four primary cartesian directions and a stay-in-place action. The state representation included frogger's position and the position of the two cars.

### 5.3    Constructing an Oracle

We used a simulated oracle in the place of human feedback, because this allows us to systematically vary the parameters of feedback likelihood, $\mathcal{L}$, and consistency, $\mathcal{C}$ and test different learning settings in which human feedback is less than ideal. The oracle was created manually by a human before the experiments by hand labeling the optimal actions in each state. For states with multiple optimal actions, a small negative reward (-10) was added to the environment reward signal of the extra optimal state-action pairs to preserve the assumption that only one action be optimal in each state.

## 6    Experiments

### 6.1    A Comparison to the State of the Art

In this evaluation we compare Policy Shaping with **Advise** to the more traditional Reward Shaping, as well as recent Interactive Reinforcement Learning techniques. Knox and Stone [7, 13] tried eight different strategies for combining feedback with an environmental reward signal and they found that

|  | Ideal Case | | Reduced Consistency | | Reduced Frequency | | Moderate Case | |
|---|---|---|---|---|---|---|---|---|
|  | ($\mathcal{L}=1.0, \mathcal{C}=1.0$) | | ($\mathcal{L}=0.1, \mathcal{C}=1.0$) | | ($\mathcal{L}=1.0, \mathcal{C}=0.55$) | | ($\mathcal{L}=0.5, \mathcal{C}=0.8$) | |
|  | Pac-Man | Frogger | Pac-Man | Frogger | Pac-Man | Frogger | Pac-Man | Frogger |
| BQL + Action Biasing | $0.58 \pm 0.02$ | $0.16 \pm 0.05$ | $-0.33 \pm 0.17$ | $0.05 \pm 0.06$ | $0.16 \pm 0.04$ | $0.04 \pm 0.06$ | $\mathbf{0.25 \pm 0.04}$ | $0.09 \pm 0.06$ |
| BQL + Control Sharing | $0.34 \pm 0.03$ | $0.07 \pm 0.06$ | $-2.87 \pm 0.12$ | $-0.32 \pm 0.13$ | $0.01 \pm 0.12$ | $0.02 \pm 0.07$ | $-0.18 \pm 0.19$ | $0.01 \pm 0.07$ |
| BQL + Reward Shaping | $0.54 \pm 0.02$ | $0.11 \pm 0.07$ | $-0.47 \pm 0.30$ | $0 \pm 0.08$ | $0.14 \pm 0.04$ | $0.03 \pm 0.07$ | $0.17 \pm 0.12$ | $0.05 \pm 0.07$ |
| BQL + **Advise** | $\mathbf{0.77 \pm 0.02}$ | $\mathbf{0.45 \pm 0.04}$ | $\mathbf{-0.01 \pm 0.11}$ | $0.02 \pm 0.07$ | $\mathbf{0.21 \pm 0.05}$ | $\mathbf{0.16 \pm 0.06}$ | $0.13 \pm 0.08$ | $\mathbf{0.22 \pm 0.06}$ |

Table 1: Comparing the learning rates of BQL + **Advise** to BQL + Action Biasing, BQL + Control Sharing, and BQL + Reward Shaping for four different combinations of feedback likelihood, $\mathcal{L}$, and consistency, $\mathcal{C}$, across two domains. Each entry represents the average and standard deviation of the cumulative reward in 300 episodes, expressed as the percent of the maximum possible cumulative reward for the domain with respect to the BQL baseline. Negative values indicate performance worse than the baseline. Bold values indicate the best performance for that case.

two strategies, *Action Biasing* and *Control Sharing*, consistently produced the best results. Both of these methods use human feedback rewards to modify the policy, rather than shape the MDP reward function. Thus, they still convert human feedback to a value but recognize that the information contained in that value is policy information. As will be seen, **Advise** has similar performance to these state of the art methods, but is more robust to a noisy signal from the human and other parameter changes.

Action Biasing uses human feedback to bias the action selection mechanism of the underlying RL algorithm. Positive and negative feedback is declared a reward $r_h$, and $-r_h$, respectively. A table of values, $H[s, a]$ stores the feedback signal for $s, a$. The modified action selection mechanism is given as $\text{argmax}_a \, \hat{Q}(s, a) + B[s, a] * H[s, a]$, where $\hat{Q}(s, a)$ is an estimate of the long-term expected discounted reward for $s, a$ from BQL, and $B[s, a]$ controls the influence of feedback on learning. The value of $B[s, a]$ is incremented by a constant $b$ when feedback is received for $s, a$, and is decayed by a constant $d$ at all other time steps.

Control Sharing modifies the action selection mechanism directly with the addition of a transition between 1) the action that gains an agent the maximum known reward according to feedback, and 2) the policy produced using the original action selection method. The transition is defined as the probability $P(a = \text{argmax}_a \, H[s, a]) = min(B[s, a], 1.0)$. An agent transfers control to a feedback policy as feedback is received, and begins to switch control to the underlying RL algorithm as $B[s, a]$ decays. Although feedback is initially interpreted as a reward, Control Sharing does not use that information, and thus is unaffected if the value of $r_h$ is changed.

Reward Shaping, the traditional approach to learning from feedback, works by modifying the MDP reward. Feedback is first converted into a reward, $r_h$, or $-r_h$. The modified MDP reward function is $R'(s, a) \leftarrow R(s, a) + B[s, a] * H[s, a]$. The values to $B[s, a]$ and $H[s, a]$ are updated as above.

The parameters to each method were manually tuned before the experiments to maximize learning performance. We initialized the BQL hyperparameters to $\langle \mu_0^{s,a} = 0, \lambda^{s,a} = 0.01, \alpha^{s,a} = 1000, \beta^{s,a} = 0.0000 \rangle$, which resulted in random initial Q-values. We set the BQL exploration parameter $\theta = 0.5$ for Pac-Man and $\theta = 0.0001$ for Frogger. We used a discount factor of $\gamma = 0.99$. Action Biasing, Control Sharing, and Reward Shaping used a feedback influence of $b = 1$ and a decay factor of $d = 0.001$. We set $r_h = 100$ for Action Biasing in both domains. For Reward Shaping we set $r_h = 100$ in Pac-Man and $r_h = 1$ in Frogger [2]

We compared the methods using four different combinations of feedback likelihood, $\mathcal{L}$, and consistency, $\mathcal{C}$, in Pac-Man and Frogger, for a total of eight experiments. Table 1 summarizes the quantitative results. Fig. 2 shows the learning curve for four cases.

In the ideal case of frequent and correct feedback ($\mathcal{L} = 1.0$; $\mathcal{C} = 1.0$), we see in Fig. 2 that **Advise** does much better than the other methods early in the learning process. A human reward that does not match both the feedback consistency and the domain may fail to eliminate unnecessary exploration and produce learning rates similar to or worse than the baseline. **Advise** avoided these issues by not converting feedback into a reward.

The remaining three graphs in Fig. 2 show one example from each of the non-ideal conditions that we tested: reduced feedback consistency ($\mathcal{L} = 1.0$; $\mathcal{C} = 0.55$), reduced frequency ($\mathcal{L} = 0.1$;

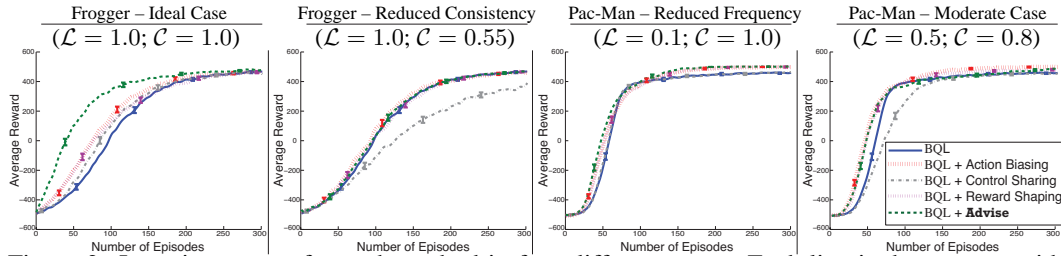

Figure 2: Learning curves for each method in four different cases. Each line is the average with standard error bars of 500 separate runs to a duration of 300 episodes. The Bayesian $Q$-learning baseline (blue) is shown for reference.

$\mathcal{C} = 1.0$), and a case that we call moderate ($\mathcal{L} = 0.5$; $\mathcal{C} = 0.8$). Action Biasing and Reward Shaping[3] performed comparably to **Advise** in two cases. Action Biasing does better than Advise in one case in part because the feedback likelihood is high enough to counter Action Biasing's overly influential feedback policy. This gives the agent an extra push toward the goal without becoming detrimental to learning (e.g., causing loops). In its current form, **Advise** makes no assumptions about the likelihood the human will provide feedback.

The cumulative reward numbers in Table 1 show that **Advise** always performed near or above the BQL baseline, which indicates robustness to reduced feedback frequency and consistency. In contrast, Action Biasing, Control Sharing, and Reward Shaping blocked learning progress in several cases with reduced consistency (the most extreme example is seen in column 3 of Table 1). Control Sharing performed worse than the baseline in three cases. Action Biasing and Reward Shaping both performed worse than the baseline in one case.

Thus having a prior estimate of the feedback consistency (the value of $\mathcal{C}$) allows **Advise** to balance what it learns from the human appropriately with its own learned policy. We could have provided the known value of $\mathcal{C}$ to the other methods, but doing so would not have helped set $r_h$, $b$, or $d$. These parameters had to be tuned since they only slightly correspond to $\mathcal{C}$. We manually selected their values in the ideal case, and then used these same settings for the other cases. However, different values for $r_h$, $b$, and $d$ may produce better results in the cases with reduced $\mathcal{L}$ or $\mathcal{C}$. We tested this in our next experiment.

## 6.2 How The Reward Parameter Affects Action Biasing

In contrast to **Advise**, Action Biasing and Control Sharing do not use an explicit model of the feedback consistency. The optimal values to $r_h$, $b$, and $d$ for learning with consistent feedback may be the wrong values to use for learning with inconsistent feedback. Here, we test how Action Biasing performed with a range of values for $r_h$ for the case of moderate feedback ($\mathcal{L} = 0.5$ and $\mathcal{C} = 0.8$), and for the case of reduced consistency ($\mathcal{L} = 1.0$ and $\mathcal{C} = 0.55$). Control Sharing was left out of this evaluation because changing $r_h$ did not affect its learning rate. Reward Shaping was left out of this evaluation due to the problems mentioned in Section 6.1. The conversion from feedback into reward was set to either $r_h = 500$ or $1000$. Using $r_h = 0$ is equivalent to the BQL baseline.

The results in Fig. 3 show that a large value for $r_h$ is appropriate for more consistent feedback; a small value for $r_h$ is best for reduced consistency. This is clear in Pac-Man when a reward of $r_h = 1000$ led to better-than-baseline learning performance in the moderate feedback case, but decreased learning rates dramatically below the baseline in the reduced consistency case. A reward of zero produced the best results in the reduced consistency case. Therefore, $r_h$ depends on feedback consistency.

This experiment also shows that the best value for $r_h$ is somewhat robust to a slightly reduced consistency. A value of either $r = 500$ or $1000$, in addition to $r = 100$ (see Fig. 2.d), can produce good results with moderate feedback in both Pac-Man and Frogger. The use of a human influence parameter $B[s, a]$ to modulate the value for $r_h$ is presumably meant to help make Action Biasing more robust to reduced consistency. The value for $B[s, a]$ is, however, increased by $b$ whenever

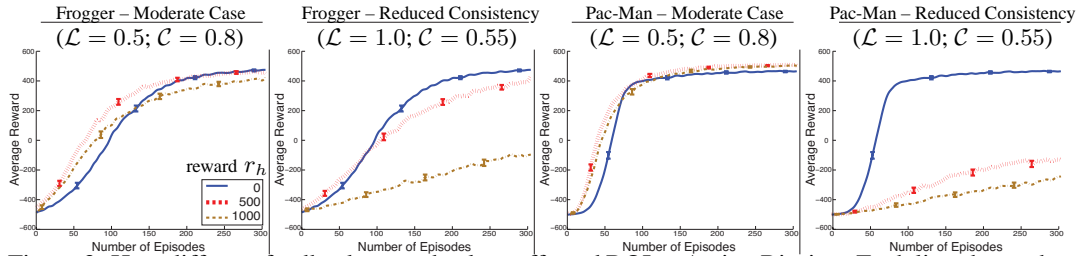

Figure 3: How different feedback reward values affected BQL + Action Biasing. Each line shows the average and standard error of 500 learning curves over a duration of 300 episodes. Reward values of $r_h = 0$, 500, and 1000 were used for the experiments. Results were computed for the moderate feedback case ($\mathcal{L} = 0.5$; $\mathcal{C} = 0.8$) and the reduced consistency case ($\mathcal{L} = 1.0$; $\mathcal{C} = 0.55$).

feedback is received, and reduced by $d$ over time; $b$ and $d$ are more a function of the domain than the information in accumulated feedback. Our next experiment demonstrates why this is bad for IRL.

### 6.3 How Domain Size Affects Learning

Action Biasing, Control Sharing, and Reward Shaping use a 'human influence' parameter, $B[s, a]$, that is a function of the domain size more than the amount of information in accumulated feedback. To show this we held constant the parameter values and tested how the algorithms performed in a larger domain. Frogger was increased to a $6 \times 6$ grid with four cars (see Fig. 4). An oracle was created automatically by running BQL to 50,000 episodes 500 times, and then for each state choosing the action with the highest value. The oracle provided moderate feedback ($\mathcal{L} = 0.5$; $\mathcal{C} = 0.8$) for the 33360 different states that were identified in this process.

Figure 4 shows the results. Whereas **Advise** still has a learning curve above the BQL baseline (as it did in the smaller Frogger domain; see the last column in Table. 1), Action Biasing, Control Sharing, and Reward Shaping all had a negligible effect on learning, performing very similar to the BQL baseline. In order for those methods to perform as well as they did with the smaller version of Frogger, the value for $B[s, a]$ needs to be set higher and decayed more slowly by manually finding new values for $b$ and $d$. Thus, like $r_h$, the optimal values to $b$ and $d$ are dependent on both the domain and the quality of feedback. In contrast, the estimated feedback consistency, $\hat{\mathcal{C}}$, used by **Advise** only depends on the true feedback consistency, $\mathcal{C}$. For comparison, we next show how sensitive **Advise** is to a suboptimal estimate of $\mathcal{C}$.

### 6.4 Using an Inaccurate Estimate of Feedback Consistency

Interactions with a real human will mean that in most cases **Advise** will not have an exact estimate, $\hat{\mathcal{C}}$, of the true feedback consistency, $\mathcal{C}$. It is presumably possible to identify a value for $\hat{\mathcal{C}}$ that is close to the true value. Any deviation from the true value, however, may be detrimental to learning. This experiment shows how an inaccurate estimate of $\mathcal{C}$ affected the learning rate of **Advise**. Feedback was generated with likelihood $\mathcal{L} = 0.5$ and a true consistency of $\mathcal{C} = 0.8$. The estimated consistency was either $\hat{\mathcal{C}} = 1.0$, 0.8, or 0.55.

The results are shown in Fig. 5. In both Pac-Man and Frogger using $\hat{\mathcal{C}} = 0.55$ reduced the effectiveness of **Advise**. The learning curves are similar to the baseline BQL learning curves because using an estimate of $\mathcal{C}$ near 0.5 is equivalent to not using feedback at all. In general, values for $\hat{\mathcal{C}}$ below $\mathcal{C}$ decreased the possible gains from feedback. In contrast, using an overestimate of $\mathcal{C}$ boosted learning rates for these particular domains and case of feedback quality. In general, however, overestimating $\mathcal{C}$ can lead to a suboptimal policy especially if feedback is provided very infrequently. Therefore, it is desirable to use $\hat{\mathcal{C}}$ as the closest overestimate of its true value, $\mathcal{C}$, as possible.

## 7  Discussion

Overall, our experiments indicate that it is useful to interpret feedback as a direct comment on the optimality of an action, without converting it into a reward or a value. **Advise** was able to outperform tuned versions of Action Biasing, Control Sharing, and Reward Shaping. The performance of Action Biasing and Control Sharing was not as good as **Advise** in many cases (as shown in Table 1) because they use feedback as policy information only after it has been converted into a reward.

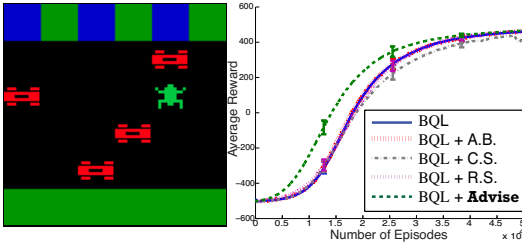

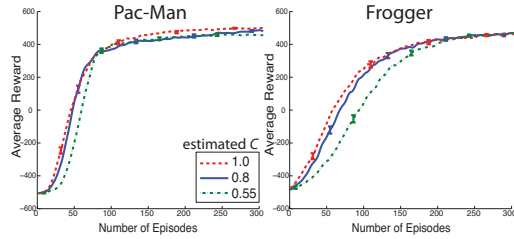

Figure 4: The larger Frogger domain and the corresponding learning results for the case of moderate feedback ($\mathcal{L} = 0.5$; $\mathcal{C} = 0.8$). Each line shows the average and standard error of 160 learning curves over a duration of 50,000 episodes.

Figure 5: The affect of over and underestimating the true feedback consistency, $\mathcal{C}$, on BQL + **Advise** in the case of moderate feedback ($\mathcal{L} = 0.5$, $\mathcal{C} = 0.8$). A line shows the average and standard error of 500 learning curves over a duration of 300 episodes.

Action Biasing, Control Sharing, and Reward Shaping suffer because their use of 'human influence' parameters is disconnected from the amount of information in the accumulated feedback. Although $b$ and $d$ were empirically optimized before the experiments, the optimal values of those parameters are dependent on the convergence time of the underlying RL algorithm. If the size of the domain increased, for example, $\mathrm{B}[s,a]$ would have to be decayed more slowly because the number of episodes required for BQL to converge would increase. Otherwise Action Biasing, Control Sharing, and Reward Shaping would have a negligible affect on learning. Control Sharing is especially sensitive to how well the value of the feedback influence parameter, $\mathrm{B}[s,a]$, approximates the amount of information in both policies. Its performance bottomed out in some cases with infrequent and inconsistent feedback because $\mathrm{B}[s,a]$ overestimated the amount of information in the feedback policy. However, even if $\mathrm{B}[s,a]$ is set in proportion to the exact probability of the correctness of each policy (i.e., calculated using **Advise**), Control Sharing does not allow an agent to simultaneously utilize information from both sources.

**Advise** has only one input parameter, the estimated feedback consistency, $\hat{\mathcal{C}}$, in contrast to three. $\hat{\mathcal{C}}$ is a fundamental parameter that depends only on the true feedback consistency, $\mathcal{C}$, and does not change if the domain size is increased. When it has the right value for $\hat{\mathcal{C}}$, **Advise** represents the exact amount of information in the accumulated feedback in each state, and then combines it with the BQL policy using an amount of influence equivalent to the amount of information in each policy. These advantages help make **Advise** robust to infrequent and inconsistent feedback, and fair well with an inaccurate estimate of $\mathcal{C}$.

A primary direction for future work is to investigate how to estimate $\hat{\mathcal{C}}$ during learning. That is, a static model of $\mathcal{C}$ may be insufficient for learning from real humans. An alternative approach is to compute $\hat{\mathcal{C}}$ online as a human interacts with an agent. We are also interested in addressing other aspects of human feedback like errors in credit assignment. A good place to start is the approach described in [13] which is based on using gamma distributions. Another direction is to investigate **Advise** for knowledge transfer in a sequence of reinforcement learning tasks (*cf.* [24]). With these extensions, **Advise** may be especially suitable for learning from humans in real-world settings.

# 8 Conclusion

This paper defined the Policy Shaping paradigm for integrating feedback with Reinforcement Learning. We introduced **Advise**, which tries to maximize the utility of feedback using a Bayesian approach to learning. **Advise** produced results on par with or better than the current state of the art Interactive Reinforcement Learning techniques, showed where those approaches fail while **Advise** is unaffected, and it demonstrated robustness to infrequent and inconsistent feedback. With these advancements this paper may help to make learning from human feedback an increasingly viable option for intelligent systems.

### Acknowledgments

The first author was partly supported by a National Science Foundation Graduate Research Fellowship. This research is funded by the Office of Naval Research under grant N00014-14-1-0003.

## Footnotes

[1]Note that the consistency of feedback is not the same as the human's or the agent's confidence the feedback is correct.

[2] We used the conversion $r_h = 1, 10, 100$, or $1000$ that maximized MDP reward in the ideal case to also evaluate the three cases of non-ideal feedback.

[3]The results with Reward Shaping are misleading because it can end up in infinite loops when feedback is infrequent or inconsistent with the optimal policy. In frogger we had this problem for $r_h > 1.0$, which forced us to use $r_h = 1.0$. This was not a problem in Pac-Man because the ghost can drive Pac-Man around the map; instead of roaming the map on its own Pac-Man oscillated between adjacent cells until the ghost approached.

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
