[Supplementary Material]

# A Appendix

Here we derive analytical solutions to the optimal feedback policy. The form of the optimal feedback policy depends on how many optimal actions a human intends to communicate per state. There are up to $2^n$ different models of the optimal feedback policy if any number of the $n$ actions can be optimal. Assuming a human will communicate fewer than $n$ optimal actions per state reduces the number of models we have to consider and thus allows us to extract more information from each instance of feedback. Here we consider the general case and the case with one optimal action per state.

## A.1 General Case

The probability an action, $a$, in state $s$ is optimal can be computed independently of the feedback to the other actions when no assumptions are made about the number of optimal actions a human intends to communicate. For brevity, let $\pi^1$ be a model of the optimal policy, $\pi^*$, in which $\pi^*(s, a) = 1$. Let $\pi^2$ be a model of the optimal policy in which $\pi^*(s, a) = 0$. The binomial distribution for the data given $\pi^1$ is: $p(d_{s,a}|\pi^1) = \binom{u_{s,a}+v_{s,a}}{u_{s,a}}\mathcal{C}^{u_{s,a}}(1-\mathcal{C})^{v_{s,a}}$, where $u_{s,a}$ and $v_{s,a}$ refer to the number of "right" and "wrong" labels respectively and $d_{s,a} = u_{s,a}, v_{s,a}$ is the feedback received for $s, a$. The binomial distribution for the data given the alternative policy is: $p(d_{s,a}|\pi^2) = \binom{u_{s,a}+v_{s,a}}{v_{s,a}}\mathcal{C}^{v_{s,a}}(1-\mathcal{C})^{u_{s,a}}$. From Bayes' rule, the probability $s, a$ is optimal is:

$$p(\pi^1|d_{s,a}) = \frac{p(d_{s,a}|\pi^1)p(\pi^1)}{p(d_{s,a})}$$

$$= \frac{p(d_{s,a}|\pi^1)p(\pi^1)}{p(d_{s,a}|\pi^1)p(\pi^1) + p(d_{s,a}|\pi^2)p(\pi^2)}.$$

The priors can be eliminated if we assume the priors have uniform probability:

$$p(\pi^1|d_{s,a}) = \frac{p(d_{s,a}|\pi^1)}{p(d_{s,a}|\pi^1) + p(d_{s,a}|\pi^2)}$$

$$= \frac{\binom{u_{s,a}+v_{s,a}}{u_{s,a}}\mathcal{C}^{u_{s,a}}(1-\mathcal{C})^{v_{s,a}}}{\binom{u_{s,a}+v_{s,a}}{u_{s,a}}\mathcal{C}^{u_{s,a}}(1-\mathcal{C})^{v_{s,a}} + \binom{u_{s,a}+v_{s,a}}{v_{s,a}}\mathcal{C}^{v_{s,a}}(1-\mathcal{C})^{u_{s,a}}}.$$

The binomial coefficients cancel (due the symmetry rule for binomial coefficients). Also, for brevity let $\Delta_{s,a} = u_{s,a} - v_{s,a}$:

$$p(\pi^1|d_{s,a}) = \frac{\mathcal{C}^{u_{s,a}}(1-\mathcal{C})^{v_{s,a}}}{\mathcal{C}^{u_{s,a}}(1-\mathcal{C})^{v_{s,a}} + \mathcal{C}^{v_{s,a}}(1-\mathcal{C})^{u_{s,a}}}$$

$$= \frac{\mathcal{C}^{u_{s,a}-v_{s,a}}}{\mathcal{C}^{u_{s,a}-v_{s,a}} + (1-\mathcal{C})^{u_{s,a}-v_{s,a}}}$$

$$= \frac{\mathcal{C}^{\Delta_{s,a}}}{\mathcal{C}^{\Delta_{s,a}} + (1-\mathcal{C})^{\Delta_{s,a}}}. \tag{1}$$

A similar derivation for the alternative policy gives:

$$p(\pi^2|d_{s,a}) = \frac{(1-\mathcal{C})^{\Delta_{s,a}}}{\mathcal{C}^{\Delta_{s,a}} + (1-\mathcal{C})^{\Delta_{s,a}}}. \tag{2}$$

## A.2 One Optimal Action Case

Deriving an analytical solution for the case with one optimal action per state first requires us to consider how to compute the probability of a model that defines a labeling for all the actions in a state, rather than just one. With $n$ actions in a state there are $2^n$ different models to consider. For

brevity let $\pi^k = \pi_1^k, \ldots, \pi_n^k$ be one of these models, where $\pi_i^k$ specifies whether $\pi^*(s,i) = 1$ or $\pi^*(s,i) = 0$. Let $D_s = d_{s,1}, \ldots, d_{s,n}$ be the accumulated human feedback corresponding to the $n$ actions in state $s$. From Bayes rule we have:

$$p(\pi^k|D_s) = \frac{p(D_s|\pi^k)p(\pi^k)}{p(D_s)}$$

$$= \frac{p(D_s|\pi^k)p(\pi^k)}{p(D_s|\pi^1)p(\pi^1) + \cdots + p(D_s|\pi^{2^n})p(\pi^{2^n})}.$$

A uniform prior over all policies allows us to eliminate the priors:

$$p(\pi^k|D_s) = \frac{p(D_s|\pi^k)}{p(D_s|\pi^1) + \cdots + p(D_s|\pi^{2^n})}$$

$$\propto p(D_s|\pi^k)$$

$$\propto p(d_{s,1}, \cdots, d_{s,n}|\pi_1^k, \cdots, \pi_n^k).$$

Repeated applications of the chain rule, followed by variable elimination using conditional independence (see Appendix A.3), gives:

$$p(\pi^k|D_s) \propto p(d_{s,1}|\pi_1^k) \times \cdots \times p(d_{s,n}|\pi_n^k)$$

$$\propto \prod_{i=1}^{n} p(d_{s,i}|\pi_i^k). \tag{3}$$

Thus, the posterior can be computed as the product of $p(d_{s,i}|\pi_i^k)$. Let $\bar{p}(d_{s,i}|\pi_i^k) = \mathcal{C}^{u_{s,i}}(1-\mathcal{C})^{v_{s,i}}$ or $\bar{p}(d_{s,i}|h_i^k) = \mathcal{C}^{v_{s,i}}(1-\mathcal{C})^{u_{s,i}}$ as needed. Also, let $N_i = \binom{u_{s,i}+v_{s,i}}{u_{s,i}} = \binom{u_{s,i}+v_{s,i}}{v_{s,i}}$. We next isolate the binomial coefficients from our computation of $p(\pi^k|D_s)$, which, in the following step, allows us to cancel them out of the equation:

$$p(\pi^k|D_s) \propto \prod_{i=1}^{n} p(d_{s,i}|\pi_i^k)$$

$$\propto \prod_{i=1}^{n} \binom{u_{s,i}+v_{s,i}}{u_{s,i}} \mathcal{C}^{u_{s,i}}(1-\mathcal{C})^{v_{s,i}}$$

$$\propto \prod_{i=1}^{n} N_i \times \mathcal{C}^{u_{s,i}}(1-\mathcal{C})^{v_{s,i}}$$

$$\propto \prod_{i=1}^{n} N_i \prod_{i=1}^{n} \mathcal{C}^{u_{s,i}}(1-\mathcal{C})^{v_{s,i}}$$

$$\propto \prod_{i=1}^{n} N_i \prod_{i=1}^{n} \bar{p}(d_{s,i}|\pi_i^k).$$

We factor out the binomial coefficients because they appear in every term of the numerator and the denominator:

$$p(\pi^k|D_s) = \frac{\prod_{i=1}^{n} N_i \prod_{i=1}^{n} \bar{p}(d_{s,i}|\pi_i^k)}{\prod_{i=1}^{n} N_i \prod_{i=1}^{n} \bar{p}(d_{s,i}|\pi_i^1) + \cdots + \prod_{i=1}^{n} N_i \prod_{i=1}^{n} \bar{p}(d_{s,i}|\pi_i^{2^n})}$$

$$= \frac{\prod_{i=1}^{n} N_i}{\prod_{i=1}^{n} N_i} \times \frac{\prod_{i=1}^{n} \bar{p}(d_{s,i}|\pi_i^k)}{\prod_{i=1}^{n} \bar{p}(d_{s,i}|\pi_i^1) + \cdots + \prod_{i=1}^{n} \bar{p}(d_{s,i}|\pi_i^{2^n})}$$

$$= \frac{\prod_{i=1}^{n} \bar{p}(d_{s,i}|\pi_i^k)}{\prod_{i=1}^{n} \bar{p}(d_{s,i}|\pi_i^1) + \cdots + \prod_{i=1}^{n} \bar{p}(d_{s,i}|\pi_i^{2^n})}.$$

Assuming policy $\pi^k$ is the case for which all actions are optimal, $p(\pi^k|D_s)$ expands as:

$$\frac{(\mathcal{C}^{u_{s,1}}(1-\mathcal{C})^{v_{s,1}})(\mathcal{C}^{u_{s,2}}(1-\mathcal{C})^{v_{s,2}})\cdots(\mathcal{C}^{u_{s,n}}(1-\mathcal{C})^{v_{s,n}})}{((\mathcal{C}^{v_{s,1}}(1-\mathcal{C})^{u_{s,1}})\cdots(\mathcal{C}^{v_{s,n}}(1-\mathcal{C})^{u_{s,n}}))+\cdots+((\mathcal{C}^{u_{s,1}}(1-\mathcal{C})^{v_{s,1}})\cdots(\mathcal{C}^{u_{s,n}}(1-\mathcal{C})^{v_{s,n}}))}.$$

The denominator above is a summation of $2^n$ different sets of products. This is equivalent to the multiplication of the $n$ different sets of the two models case (see Appendix A.1), as shown:

$$
\begin{aligned}
p(\pi^k|D_s) &= \left(\frac{\mathcal{C}^{u_{s,1}}(1-\mathcal{C})^{v_{s,1}}}{\mathcal{C}^{u_{s,1}}(1-\mathcal{C})^{v_{s,1}}+\mathcal{C}^{v_{s,1}}(1-\mathcal{C})^{u_{s,1}}}\right)\cdots\left(\frac{\mathcal{C}^{u_{s,n}}(1-\mathcal{C})^{v_{s,n}}}{\mathcal{C}^{u_{s,n}}(1-\mathcal{C})^{v_{s,n}}+\mathcal{C}^{v_{s,n}}(1-\mathcal{C})^{u_{s,n}}}\right) \\
&= \left(\frac{\mathcal{C}^{\Delta_{s,1}}}{\mathcal{C}^{\Delta_{s,1}}+(1-\mathcal{C})^{\Delta_{s,1}}}\right)\cdots\left(\frac{\mathcal{C}^{\Delta_{s,n}}}{\mathcal{C}^{\Delta_{s,n}}+(1-\mathcal{C})^{\Delta_{s,n}}}\right) \\
&= \prod_{i=1}^{n} p(\pi_i^k|d_{s,i}). 
\end{aligned}
\tag{4}
$$

Equation (4) represents the likelihood of model $\pi^k$, which is useful if we restrict the number of possible optimal actions per state. We could use Eqn. 4 without restricting the number of optimal actions per state, but finding the model with maximum likelihood would involve comparing the likelihoods of $2^n$ different models, which is intractable. For our purposes, we limit the number of possible optimal actions per state to one. This decreases the number of models we need to consider to $n$. Let $\pi^k$ correspond to the model that action $k$ is optimal and all others are suboptimal. We have:

$$
\begin{aligned}
p(\pi^k|D_s) &\propto p(\pi_k^k|d_{s,k})\prod_{i\neq k}^{n} p(\pi_i^k|d_{s,i}) \\
&\propto \mathcal{C}^{\Delta_k}\prod_{i\neq k}^{n}(1-\mathcal{C})^{\Delta_{s,i}} \\
&\propto \mathcal{C}^{\Delta_k}(1-\mathcal{C})^{\sum_{i\neq k}^{n}\Delta_{s,i}}.
\end{aligned}
\tag{5}
$$

Equation (5) is what we use with **Advise**.

## A.3  Simplification of the probability $p(D_s|\pi^k)$

We can make the computation of $p(D_s|\pi^k)$ tractable through repeated application of the chain rule and conditional independence relations. First we expand the data variables using the chain rule:

$$
\begin{aligned}
p(D_s|\pi^k) &= p(d_{s,1},\cdots,d_{s,n}|\pi^k) \\
&= p(d_{s,1}|\pi^k)\times p(d_{s,2},\cdots,d_{s,n}|d_{s,1},\pi^k) \\
&= p(d_{s,1}|\pi^k)\times p(d_{s,2}|d_{s,1},\pi^k)\times p(d_{s,3},\cdots,d_{s,n}|d_{s,1},d_{s,2},\pi^k) \\
&= p(d_{s,1}|\pi^k)\times p(d_{s,2}|d_{s,1},\pi^k)\times\cdots\times p(d_{s,n}|d_{s,1},\cdots,d_{s,n-1},\pi^k).
\end{aligned}
$$

We next use conditional independence to eliminate the dependence among the data variables. Because we take the policy to be true, and because action selection is the same for all the models, the data received for one action does not help to explain the data received for the other actions. This allows us to eliminate the dependence among the data:

$$p(D_s|\pi^k) = p(d_{s,1}|\pi^k)\times p(d_{s,2}|\pi^k)\times p(d_{s,3}|\pi^k)\times\cdots\times p(d_{s,n}|\pi^k).$$

We can use similar reasoning to eliminate the dependence on multiple model variables:

$$
\begin{aligned}
p(D_s|\pi^k) &= p(d_{s,1}|\pi^k)\times p(d_{s,2}|\pi^k)\times\cdots\times p(d_{s,n}|\pi^k) \\
&= p(d_{s,1}|\pi_1^k,\cdots,\pi_n^k)\times p(d_{s,2}|\pi_1^k,\cdots,\pi_n^k)\times\cdots\times p(d_{s,n}|\pi_1^k,\cdots,\pi_n^k) \\
&= p(d_{s,1}|\pi_1^k)\times p(d_{s,2}|\pi_2^k)\times\cdots\times p(d_{s,n}|\pi_n^k) \\
&= \prod_{i=1}^{n} p(d_{s,i}|\pi_i^k).
\end{aligned}
$$