[Reviews · NeurIPS 2013]

Submitted by Assigned_Reviewer_4

The paper presents an approach to interactive reinforcement learning there human feddback is interpreted directly as policy labels.
The paper is clearly written and easy to understand. The method is sound and based (to my understanding) well on the existing literature.
In my opinion the papers strongest point is that the method presented (named Advise) is simple, needs less meta parameters than state of the art methods and this single meta parameter C (that depicts the estimated consistency of the human feedback) is also not very sensitiv. In combination with the results that show that Advise performs better or equal to the state of the art approaches, Advise seems to me to be an very interesting method.

But the paper has also some weaknesses, especially for a NIPS submission:
The examples that were used as benchmarks seem too easy.
Also the theoretical delta of the method to the state of the art is not very large.

Because the idea is interesting and the method itself is compelling I still tend, however, slightly to suggesting acceptance of the paper.

There are also some minor points:
Page 1, line 32 or 33 (the numbering is a bit off in the PDF): "In this paper WE introduce..."
Page 2, line 75 or 76: "This is THE most common..."
Page 5, Table 1: This table is in my opinion too small.
Pages 6-8, Figures 2-5: This figures are definitively too small (at least in the printout). I know its hard to meet the page-limit in NIPS, but the ticks are not readable and the plots themselves are too close on top of each other.
Page 7, Line 373 or 374: "interpret feedback is as a direction" - please rephrase.
Summary: The paper presents an interesting method for interactive reinforcement learning that is simpler, with less meta parameters by showing equal or better performance than state of the art methods.
It lacks however involved theoretical innovation and demonstrates the performance only on two simple benchmarks.

Submitted by Assigned_Reviewer_5

This paper presents a new method for using human feedback to improve a reinforcement-learning agent. The novelty of the approach is to transform human feedback into a potentially inconsistent estimate on the optimality of an action, instead of a reward as is often the case. The resultant algorithm outperforms the previous state of the art in a pair of toy domains. I thought this was an excellent paper, which appropriately motivated the problem, clearly introduced a new idea and then compared performance to other state-of-the-art algorithms (and not just strawmen). I mostly have suggestions for improvement.

- I really liked the use of simulated human teacher, which could be manipulated systematically to change likelihood and consistency of feedback. One thing I would have liked to see is much lower likelihoods of feedback (< 1%)

- Something that worries me is that people may be systematically inconsistent in their feedback. In psychology, one of the most common uses of reward shaping is in the process of training a new behaviour through successive approximation. That is, let’s say you want a rat to pull a chain. First, you would reward the rat for getting near the chain, then for touching it, and finally for pulling on it. At each step, you deliberately stop rewarding the earlier step. How would Advise deal with this type of systematic inconsistency in the human feedback (which is the type of feedback they might get from an expert human trainer)?

- Sec 4.2: I found the assumption that humans only know one optimal action to be a bit too strong. What happens to the algorithm if that assumption is relaxed? Is performance compromised if the human teacher vacillates between shaping two different optimal actions? Maybe it should? A few words on this issue would be nice.

- One other issue that arises in working with human feedback is delay. Much inconsistency may simple be due to people not responding at the same rate each time—i.e., giving positive feedback only after another intervening action. I think this might actually be another reason that the Advise approach (which allows for inconsistency) is stronger than the other alternatives considered.

Minor things:

line 053: “from MDP reward” is an awkward construction
sec 5.1. How do you win in Pac-Man? Eat both dots? Not specified.
Table 1 (and figures). Second column would be clearer as “Reduced Frequency” instead of “Reduced Feedback”. Also, the ordering of conditions (from left to right) is different in Table 1 than the subsequent figures.
lines 234-247. The relation between control sharing and action biasing could be made a little clearer.
lines 294. prior estimate of the (missing of)
Figure 2. Other than the ideal case, why choose to plot only those cases where Advise does not help (if I am reading Table 1 correctly)?
line 369-370. More accurately, it’s probably best to take the closest overestimate of C (i.e., err upward).
Figures 4 and 5: The text in the figures (esp the axis labels) was way too small.
Summary: This paper presents a new method for using human feedback to improve a reinforcement-learning agent. The approach was novel, and the experiments nicely showed improvement performance against other state-of-the-art approaches.

Submitted by Assigned_Reviewer_6

The paper proposes a Bayes-optimal approach for integrating human feedback with reinforcement learning. The method extends and is compared to the baseline of Bayesian Q-learning. The approach is tested experimentally on two domains (Pac-Man and Frogger).


Quality
-------
Overall, the paper is sound and logical. However, there are a few problems:

1) The authors claim that their approach does not convert the feedback to rewards or values. But, by calculating delta_{s,a} from the count of the labels (right/wrong) they essentially convert the labels to values.

2) The name of the proposed approach (policy shaping) is a bit misleading. In fact, the feedback is given per action, not per whole policy/episode during the learning. Therefore, a more appropriate name would have been, maybe, "action shaping".


Clarity
-------
The paper is generally well written and flows logically. There are a few parts, though, that are a bit confusing:

1) Section 4.2. In it, the authors at first state the assumption that the optimality of an action a in a given state s is independent of the labels provided to other actions in the same state. This leads to formula (1). However, the following formula (2) violates this assumption by relying on the values delta_{s,j} from other actions in the same state.

2) The first time the authors clearly state how the human feedback is given (as binary right/wrong labels immediately following an action) comes too late in the text (around line 112, on page 3). It should have been much earlier in the text.

3) Section 5.3. It is not entirely clear to me how the pre-recorded human demonstrations are used to produce a simulated oracle.


Originality
-----------
Unfortunately, some of the most interesting problems are left for future work (e.g. the credit assignment problem, mentioned on line 125, as well as the case when there is more than one optimal action per state).

The proposed method for resolution of the multiple sources does not seem to be elaborate enough. By multiplying the two probabilities, both of them are taken into account with equal weight, even if one of them is less reliable than the other. A better approach would have been to use the value of C to evaluate the reliability of the human feedback and take this into consideration.


Significance
------------
In my opinion, the demonstrated improvement by using the additional human feedback is not sufficiently significant to justify the large amount of additional information needed by the algorithm. In fact, if the "oracle" information is directly provided to the agent in the form of "demonstrations", the agent would be able to "jump-start" the learning from a very high-return initial policy, and further improve it during the episodes.


Summary: The paper proposes a Bayes-optimal method for inclusion of binary right/wrong action labels provided by human into reinforcement learning. The paper is well written, but could be further improved in terms of clarity, originality and significance.
Author Feedback

Author rebuttal: We thank the reviewers for their thoughtful reviews. The minor text suggestions have been implemented and will improve the clarity of our paper. Below we provide a response to other questions raised in the reviews.

R4

Re: theoretical advances. Our method is most similar in principal to state-of-the-art methods like Action Biasing. The main theoretical innovation lies in our probabilistic interpretation of human feedback, allowing us to do away with tuning parameters related to equating feedback to a reward value. This provides us with a new way to reason about policies learned from human feedback. Explicitly accounting for uncertainty in the interactive learning scenario is, we believe, an important insight that is unique to Advise.

Re: simple domains. Our benchmarks highlight the benefits of this interpretation. We designed them to focus on Advise’s robustness to parameter changes as well as its effectiveness under different feedback conditions. The two domains, though simple, show that Advise is a stable and effective approach and motivates the need to pursue research using more complex domains.

R5

Re: systematic inconsistency. In this case, Advise plays the role of guiding exploration. An instance of feedback for a specific state-action pair encourages exploration in that part of the state-action space more than others, while the agent is simultaneously performing RL value function updates to learn an estimate of the external value of different state-action pairs. Thus, similar to a rat, once the human’s “shaping” feedback subsides, it's the agent’s learned policy based on that previous shaping that should yield proper behavior.

Re: more than one optimal action. Advise can assume either a single optimal action or multiple optimal actions per state. Assuming a single optimal action per state allows an agent to extract more information from a single instance of feedback because it informs the agent about the optimality of other actions (e.g., a yes to one action is a no to other actions). Under this assumption, labeling multiple actions as optimal will still eventually make those actions more likely than the non-optimal ones, just more slowly. Similarly, in the formulation that allows for multiple optimal actions per state, feedback only modifies the uncertainty about the state-action pair it is applied to, resulting in a longer time for a single optimal action to peak in the distribution. We chose to assume that even when there are multiple optimal actions, a human would choose one far more often.

R6

Re: delta_{s,a} is a value. Advise uses delta_{s,a} as a value in the sense that it converts feedback to an estimate of the likelihood of performing a particular action in a state; however, delta_{s,a} is not used to compute a value/utility in the traditional RL sense, nor does it directly interact with the values/utilities used inside an RL algorithm. This is an important distinction.

Re: reliability. We are not sure we understand the reviewer’s question/suggestion. Our formulation, by definition, accounts for the uncertainty in the respective distributions. The value of C exactly denotes the reliability of the human feedback policy while the uncertainty in the estimated long-term expected discounted rewards in each state (in our case the precision of the normal distribution) are used to compute the reliability of the BQL policy. The probability distribution of each represents their respective uncertainties in the optimal policy for a state. The Bayes optimal approach to combine these two conditionally independent distributions is to multiply them (as shown in [21] and [22]). One effect of our approach is, for example, when the human feedback policy is unreliable (C is near 1/2), the combined policy will be nearly equivalent to the RL policy. If we misinterpret the reviewer's suggestion, we would appreciate any further clarification and will try to incorporate it into our paper.

Re: using demonstrations. Using human demonstrations from start to end is a form of learning from human feedback; however, we wanted to explore the state-by-state feedback case because: it can sometimes be easier for someone to critique a policy on the fly rather than to provide a complete and optimal demonstration; humans are not necessarily confident about what to do everywhere; and, we wanted to explore how one might benefit even from sparse feedback. Certainly, the two approaches for making use of human feedback can be mutually beneficial. In fact, one can convert a complete trajectory into state-by-state feedback to take advantage of our approach directly.